# Current Epidemic Situation and Control Status of Citrus Huanglongbing in Guangdong China: The Space–Time Pattern Analysis of Specific Orchards

**DOI:** 10.3390/life13030749

**Published:** 2023-03-10

**Authors:** Jingtian Zhang, Yangyang Liu, Jie Gao, Chunfen Yuan, Xuanlin Zhan, Xiaoqing Cui, Zheng Zheng, Xiaoling Deng, Meirong Xu

**Affiliations:** 1Guangdong Province Key Laboratory of Microbial Signals and Disease Control, Citrus Huanglongbing Research Laboratory, South China Agricultural University, Guangzhou 510642, China; 2Department of Entomology, China Agricultural University, Beijing 100193, China

**Keywords:** surveillance, *Candidatus* Liberibacter asiaticus, comprehensive control, distribution, screen house

## Abstract

Huanglongbing (HLB) is the most harmful bacterial disease in citrus production in the world, and has been seriously ravaging the citrus groves of South China since the 1930s. The surveillance of the epidemiological characteristics of HLB is of utmost priority for citrus production in this region. In order to explore the effects of disease control measures, analyses on the space–time statistical features of the HLB epidemic, from 2019 to 2021, within six orchards in the Guangdong province are presented. Overall, the number of citrus plants in the orchards usually slightly decreased year by year. The reduction was mainly related to the level of plant susceptibility, which is correlated with citrus varieties. The maximum disease severity (incidence and race increment) was correlated with the awareness of this disease and the management intensity applied by the manager. A higher disease index was found in the conventional management orchards than in the comprehensive prevention and control orchards. Proper insect-protective screen houses can effectively prevent the epidemic of HLB, without affecting the fruit quality, and can also aid with higher yields. A high correlation was found between the geometry and topography of orchards and the HLB epidemic due to the wind direction from May to September and the Asia citrus psyllid activity characteristics. For flat orchards, the incidence of HLB in the north and entrance areas was higher than that in the southwest. In the mountain area, the incidence of the windward side in the south was higher than that of the leeward side in the north. Diseased trees tended to have an edge effect in the grove, whereas the trees of the same disease scale were found clustered in their distribution. These results allow a better understanding of HLB epidemiology and provide guidance for the early warning of HLB in new groves in areas that are severely affected by this disease. Furthermore, they also provide a scientific basis for the comprehensive prevention and control of HLB in old groves.

## 1. Introduction

Citrus, including oranges, tangerines, grapefruits, pomelos, lemons, limes, etc., is a perennial grafted crop of the most planted and produced fruits in the world (https://www.statista.com, accessed on 5 January 2023). Moreover, citrus Huanglongbing (HLB) is one of the most crucial and devastating diseases in the citrus industry worldwide. It was included as one of the top ten crop diseases by the Ministry of Agriculture and Rural Affairs of China in 2020. HLB causes the yellowing of new shoots, the mottling of fully mature old leaves, or a zinc-deficiency-like symptom in mature fresh leaves; however, these symptoms occur with no programmed cell death (PCD), such as necrosis. Among these, the mottling leaf is the most typical symptom to identify HLB. Compared to normal healthy fruits, the most reliable diagnostic symptoms of affected fruits are being improperly colored (greening or ‘red nose fruit’), being lopsided with a curved columella, and developing mostly aborted seeds. These phenomena of the various symptoms mentioned are particularly prevalent in Guangdong, as HLB ravages the citrus industry there [1,2,3].

Huanglongbing is caused by phloem-limited bacteria, specifically, the ‘*Candidatus* Liberibacter spp.’ of α-proteobacterium [4]. Three distinct Liberibacter species are related to HLB. The HLB caused by ‘*Candidatus* Liberibacter asiaticus’ (CLas) have a wider geographical spread, are more severe, present in lower elevations, and possess higher temperature tolerance [5]. Further, it is vectored by Asia citrus psyllid (*Diaphorina citri* (Kuwayama)) [6] in a persistent and propagative manner. The HLB caused by ‘*Ca*. L. africanus’ (CLaf) and ‘*Ca*. L. americanus’ (Clam) are more restricted, less severe, and more temperature-sensitive [7]. CLas is the only species detected from HLB-affected citrus in China [8]. After infection, CLas quickly colonizes the root system before canopy symptoms develop. The upward movement of CLas from roots to canopy is linked to seasonal flushes and the CLas population [9]. Globally, HLB has been distributed in 64 countries within Asia, Africa, North America, South America, and Oceania (data accessed on 20 December 2021 from Center for Agriculture and Bioscience International (CABI)). The pathogen, CLas, is widely distributed in Asia and North America, and partially distributed in Africa [10,11,12]. Though CLam is only partially detected in Brazil [13], there has been a shift in the prevalence of CLam to CLas, which has been recently observed [7].

Citrus cultivation was highly developed early last century in the Pearl River Delta and the Hanjiang Delta in the Guangdong province of China [14]. As determined from the tonnage data of citrus production, the citrus industry peaked in 1933; however, a significant drop occurred in 1934 in several areas, including the Chaoshan area in Hanjiang Delta. According to Lin Kong-Hsiang’s publications [1], the HLB epidemic in the Chaoshan area could be traced back to as early as the 1870s, despite its, then, slight prevalence status. Whereas the first report of this disease was most likely recorded by Reinking [15] after a disease survey along the Pearl River Delta and West River, it became a serious disease from the 1930s onwards in Guangdong, with the citrus production cycle becoming shorter as a result [1,14,15,16,17,18,19]. The report and incidence of HLB in the Fujian and Guangxi provinces was a little bit later than those found in Guangdong. Nonetheless, the HLB in these three provinces has been considered the most widely distributed and the most serious since the mid-20th century. From the late 1970s until the 1980s, HLB became a serious issue in Sichuan, Jiangxi, Yunnan, Hainan, and Taiwan. It then spread into Hunan, Guizhou, and Zhejiang [19] after 1980. Until now, 11 of the 19 citrus cultivation provinces and regions in China, accounting for more than 80% of the total citrus cultivation area, had been damaged by HLB [19].

In recent years, the damage caused by HLB in the Guangdong Province has been increasing; further, it is most likely that the shortened citrus production cycle from planting to replant in the whole orchard is a reflection of this fact. The citrus producers summarized a famous ‘ten-year cycle’, which means that the trees in the orchards would be replaced every 10 years. This would be performed because most trees were affected and, thus, were producing limited valuable fruits. Take the famous Yangcun citrus farm in Huizhou, which is the largest (2000 hm^2^) citrus farm in Asia and was established in 1951–1953, as an example. The farm experienced two large-scale disease tree eradications from 1979 to 1982 and during 1996–1999. The production of the fresh fruits reached its peak in 1977 and 1991, but dropped significantly in 1982 and 2000 [17]. Since the ravages of HLB from the 1970s, the citrus production in this area cannot avoid the ‘ten-year cycle’. After that, the most famous local varieties, e.g., the *Citrus reticulata* ‘Shatang’ tangerine in Sihui, the *C. sinensis* Osbeck cv. ‘Hongjiang’ orange in Zhanjiang, and the *C. reticulata* Blanco ‘Tankan’ in Puning, were also almost destroyed at the beginning of this century. A recent survey in 2016–2017 concluded that citrus groves in all 17 cities and 65 counties in the Guangdong Province were affected by HLB, and around 59,700 hectares of citrus were affected, accounting for about 25% of the total citrus planting area [20]. Currently, HLB still seriously restricts the development of the citrus industry in Guangdong. However, with more attention being paid recently via government policy toward the comprehensive prevention and control of HLB, the resumption of citrus production has been promoted. More growers prefer large-scale groves, where standardized management with higher technical specifications is adopted.

Recorded systematic investigation on the occurrence of HLB in the field were mainly concentrated around the 1950s and 1980s [17]. Most of those studies focused on the annual increase and removal of diseased trees in orchards. A few of these investigations analyzed the correlation between geographical environment conditions and the prevalence of HLB. Quite a few studies have analyzed the relationships between HLB prevalence results, and cultivation and management measures. With the development of molecular biology technology, after the pathogen of HLB was identified as a type of bacteria [4], the research hotspots have shifted from disease epidemic to the molecular interaction among the pathogen–host–vector. Although HLB has been ravaging the citrus industry of Guangdong Province continuously, no studies have systematically investigated the annual prevalence of HLB within specific orchards in the last 30 years.

In recent decades, mathematical models play an important role in understanding the epidemiology of HLB [21,22,23,24]. However, most of the dynamic behaviors of these models are studied by only using computer simulations or are only understood by professional persons. In this study, space–time dynamic (year-by-year) point pattern measures were applied to highlight the HLB progression over time in the groves of Guangdong. All the screened orchards were larger than 5 hectares and with different management levels. A ‘two-to- three year’ survey was conducted for the appearance and degree of HLB symptoms on each tree within these groves for the appearance and degree of HLB symptoms. In this study, the aims are to provide guidance for the early warning of HLB, to enhance the prevention awareness of growers, and to provide suggestions on the specific effective measures for the prevention and control of HLB in areas where HLB is severely endemic.

## 2. Materials and Methods

### 2.1. Orchard Information and Survey Methods

To reduce the impact of spatial heterogeneity, the research area is located in South China, specifically in the province of Guangdong. Six orchards in Zhaoqing City, Huizhou City, Guangzhou City, or Meizhou City were selected as the survey sites. Only one Liberibacter species (CLas) was detected in these areas. The tree density of the orchards was similar in all orchards, except for orchard 1, wherein the cultivar within it was usually planted at a high density. The information on the selected orchards is shown in Table 1. The A1 region of orchard 4 was under semi-natural conditions in a screen house. Orchard 5 and orchard 6 contain both sloping plots and non-sloping plots. In the conventional management orchards, the diseased trees were not timely monitored or rouged, and attention was not paid to killing the psyllid. Comprehensive control is taken to mean integrated management concepts, which combine cultural, chemical, and biological control measures that are conducted between 9 and 12 times in a timely manner with pesticide application.

Investigations were carried out via visual inspection in Autumn and Winter from 2019 to 2021, when the diseased citrus varieties showed their apparent symptoms (Figure A1). During each survey, 12 experienced Citrus Huanglongbing Research Laboratory members participated, with 2 persons in a group. Each investigator collected at least 20 leaf samples which were visually inspected as HLB–affected. Further, another 20 samples evaluated as healthy were collected each day. DNA was extracted from the randomly collected samples and amplified by RT-PCR for CLas detection [25]. If the PCR results of 95% of the collected samples were confirmed to be consistent with the visual inspection, the data from the survey were used for further analysis. Otherwise, the surveyed plot would be assigned to another pair of investigators. In addition, the CLas-positive samples were assigned for genetic diversity analysis by conventional PCR and RT-PCR based on the phage types (three type-specific prophage loci), a miniature inverted-repeat transposable element (MITE) region (CLIBASIA_05620 ~ CLIBASIA_05625), and their short tandem repeat genes (CLIBASIA_03080 and CLIBASIA_01215) [26]. After collecting the band information of conventional PCR or Ct values of RT-PCR according to Zheng et al. [26], data were further used for diversity analysis of the CLas strains. Cluster analysis of the CLas populations in the six orchards was completed by Popgene v. 1.32 (https://www.ualberta.ca/~fyeh/popgene.html, accessed on 3 March 2023) based on Nei’s (1972) genetic distance. The cluster map was generated in MEGA v. 11.0 [27].

The scales (0, 1, 2, and 3) of the diseased trees were recorded based on the severity of the disease, wherein scale 0 indicates non-HLB-affected trees and scale 1 indicates that 1/3 of the canopy is affected by HLB, etc. The ‘two-step-path’ app (Shenzhen 2bulu Information Technology Co., Ltd., Shenzhen, China) was applied to record the path/track of the survey, to map the diseased trees in space and to record the time of the surveys. The data from the ‘two-step-path’ app were viewed and exported by a LocaSpaceViewer4 PC (Zhongke Tuxin Technology Co., Ltd., Suzhou, China). These procedures resulted in the development of space–time point pattern survey maps of the different scaled symptomatic citrus. Counting was carried out in each survey for the total number of trees, the number of those which were removed, and the number of infected trees. The incidence of the latter three was calculated by normalizing the counts to the total number of plants in the orchard.

### 2.2. DNA Extraction and RT-PCR

The midribs of collected leaf samples were cut into small pieces with sterilized blades. In addition, 0.05 g of them were ground by a FastPrep tissue homogenizer (MP Biomedicals, Irvine, CA, USA). A subsequent extraction of DNA was performed using the E. Z. N. A. HP Plant DNA Kit (Omega Bio-tek., Norcross, GA, USA), according to the manufacturer’s protocol. The concentration and purity of the DNA samples were determined using a NanoDrop™ One spectrophotometer (Thermo Fisher Scientific, Shanghai, China).

The RT-PCR assays for the CLas detection were performed with a primer set (CLas4G/HLBr) and a probe (HLBp) according to a previous study of our lab [28]. The 20 μL of PCR mixture contained 1 μL of a DNA template (~25 ng), 10 μL of a Bestar qPCR Master Mix (DBI Bioscience, Shanghai, China), 0.2 μL of PCR Probe (10 μM), 0.4 μL of each forward and reverse primer (10 μM), and 8 μL of DNase/RNase-Free ddH_2_O. Standard TaqMan thermocycling conditions were used: 95 °C for 2 min, followed by 40 cycles of 95 °C for 10 s and 58 °C for 30 s, with fluorescence signal captured at the end of each 58 °C step. All PCR assays were run in triplicate in a CFX Connect Real-Time System (Bio-Rad, Hercules, CA, USA). The data were analyzed using Bio-Rad CFX Manager 2.1 software with the automated baseline settings and threshold.

### 2.3. Measurement of Citrus Canopy, Yield, and Juice Quality

In order to assess the effects of a protective screen on the growth of citrus plants, the A1 and A2 areas in orchard 4, with and without the coverage of an insect-proof screen, were divided into 5 blocks (i.e., south, west, north, east, and center). Two healthy and two HLB-affected trees were selected, respectively, from each block of the two regions. The height and crown diameter were measured monthly using a tape, for a year. The height measurement of each tree was performed three times in three directions, i.e., the south–north, 45° to the south–east, and 45° to the south–north. The diameter of each tree was represented by the average data extracted from the two measurements in the directions of south–north and east–west. The crown surface area (S) was estimated via using the formula of S = 4πR^2^. In the formula, R was calculated by the average height and diameter.

To analyze the influences of the HLB and the semi-natural cultivation model on the yield, a total of 20 healthy trees and 20 diseased trees were selected from different blocks. Fruits from every selected tree were collected and weighted with ten replications. As the HLB symptoms are often sectored within a tree, only visually symptomatic fruits from the diseased trees were selected for fruit quality assessment. The following morphological characterizations were conducted: single fruit weight (FW); the fruit transverse and longitudinal diameters (FTD and FLD); outer pericarp thickness (OPT); outer pericarp weight (OPW); fruit shape index (FSI); fruit firmness (FF); concentration of vitamin C; total soluble solids content (TSS); and the total titratable acids (TA). The FSI was calculated by the following formula: FSI = FTD/fruit surface area × 100. Puncture and compression tests were based on a texture analyzer, which were used to assess FF. The fruit mass rate and outer pericarp rate were also obtained.

Collected fruits were transported to the laboratory and temporarily stored at 4 °C. After peeling, the hand-pressed juice was filtered through four layers of sterile gauze pieces, and then collected into sterile containers. The content of vitamin C was measured in the freshly squeezed juices via a 2,6-dichlorophenol indophenol redox titration method [29]. The TSS and TA were determined by a sugar acid digital display refractometer (PaL-BXIACID F5, Atago Co, Tokyo, Japan).

The color values of fruits on the surface were evaluated using a chromameter tristimulus color analyzer, which was calibrated with a white porcelain reference plate. Each fruit was measured six times in the upper, middle, and lower parts. The apparatus calculated and returned the three color parameters from the spectra. The color coordinates of the uniformed color space L, a, b, and hue angle (H°) were determined [30]. The L values (ranging from 0.09 to 107.26) represent the luminosity. Both a and b values represent different colors, with a values ranging from −60 (green) to 60 (red), whereas b values range from −60 (blue) to 60 (yellow). The H° = hab = arctg (b/a) is the qualitative attribute that graded any color as reddish, greenish, etc.

### 2.4. Statistical Analyses

The morphological characterization (FW, FTD, FLD, OPT, etc.) and the quality data analyzed were collected and averaged using Microsoft Excel software (Microsoft, Redmond, Washington, DC, USA). The minimum, maximum, and mean values; the standard deviations (SD); and the coefficient of variation (CV) were calculated separately for the measured traits among the individuals of the different groups’ fruits. F-test, one-way ANOVA, and S Shapiro–Wilk tests were performed for the relevant data sets. To test whether the traits of the healthy and diseased fruits, both inside and outside the screen house, differed significantly, we ran independent-sample *t*-tests. A one-way analysis of variance (ANOVA) test was used to determine the significant differences in the measured traits. A Pearson correlation coefficient was then used to determine the relationships between the traits and the infection status or the screen house coverage. The analysis of variances between the two different experimental groups was conducted with Tukey’s post hoc comparison test. The data were analyzed using SPSS 19.0 (SPSS Inc., Chicago, IL, USA). The bar chart was generated using the software of Origin 2021 (OriginLab Corp., Northampton, MA, USA).

## 3. Results

### 3.1. The Epidemic Characteristics of HLB Were Correlated with Management Level

All collected DNA extracted from six orchards’ CLas–positive samples were used to analyze the prophage types, short tandem repeat genes, and MITE region by conventional PCR or RT-PCR. The clustering result of these CLas strains is shown in Figure 1. The CLas population from orchard 5 (Guangzhou, China) was different from populations of other orchards, whereas the CLas populations collected from orchard 4 (Huizhou, China) and orchard 6 (Meizhou, China) were similar. Similarly, the CLas populations from orchard 1, 2, and 3 in Deqing county were also highly similar.

Of the six surveyed orchards, two were under conventional management, and three received comprehensive prevention control practices. The number of citrus plants decreased yearly in all orchards except for orchard 3. This reduction was found related to the severity of symptoms caused by HLB rather than to the management level. For example, after being affected by HLB, *C. medica* ‘Fingered’ and *C. maxima* (Burm.) Merr. will undergo a long asymptomatic stage and subsequently create a hidden epidemic of this disease. Thus, in three years, the number of citrus plants in the two orchards reduced from 17,964 to 17,447, and from 5893 to 5748. By contrast, the diseased trees of *C. reticulata* Blanco ‘Shatangju’ (from 7032 to 5277), *C. reticulata* Blanco var. ‘Gonggan’ (from 5326 to 4205), and *C. reticulata* Blanco ‘Wokan’ (from 15,171 to 14,065) were eradicated mainly due to remarkable symptoms. The affected trees of these three cultivars evidently declined in Autumn and Winter, with the leaves yellowing and falling off, and the fruits being small and deformed. The epidemic characteristics of HLB in the two ‘Gonggan’ orchards under conventional management (orchard 3) and comprehensive control (orchard 4) in Deqing were subsequently compared. The disease incidence rates of these two orchards differed distinctly (Figure 2). The HLB rate at the former farm rose from 15.58% (830/5326) to 30.65% (1289/4205), whereas that of the latter rose from 0% to 0.18% (21/11,623). Due to the conventional methods implemented in the former orchard, the population of *D. citri* was observed to be at a stable level during the year. Unfortunately, the owner did not have the intensive awareness to prevent and control the psyllid. As a result, the orchard was fully replanted in 2022. The latter orchard was managed by a famous company. The location of the grove was carefully selected, and the nursery stocks at the beginning of the orchard establishment were strictly inspected as to whether they were free of CLas and other viruses. Additionally, cultivation management in the later orchard was carried out according to certain technical regulations for the prevention and control of HLB (T/SHSTJ 002—2020).

For the other three orchards (orchards 1, 4, and 5), which possessed a certain HLB incidence base (2–6.5%) in 2019, the disease epidemic situations differed during the investigation period (Figure 2). From 2019 to 2021, the incidence of the fingered citron orchard in Deqing (orchard 1) increased from 6.49% (1165/17,964) to 15.78% (2753/17,447); it was finally destroyed due to the damage of HLB and freezing. By contrast, the incidence of the comprehensive control demonstration orchard in Yangcun (orchard 4) decreased from 6.06% (436/7032) to 3.22% (220/6834), and then increased to 9.51% (502/5277) from 2019 to 2021. In this orchard, the sudden HLB incidence increase was related to the removal of all pomelos trees that were planted at the edge of the orchard. The *D. citri* could easily invade inward, thus leading to the quick spread of HLB; this phenomenon is called the ‘marginal effect’. For the comprehensive control orchard in Conghua, the 3-year incidence of HLB was 2.13% (323/15,170), 0.94% (142/15,171), and 3.29% (478/14,065), respectively. Chemical and physical methods were used to control *D. citri* in the two comprehensive control orchards from 2019 to 2020. However, instead of chemical control, the Conghua orchard began to use biological control methods in 2021, which led to an increase in the *D. citri* population and to a spread of HLB.

### 3.2. Screen Houses Effectively Prevent the HLB Epidemic without Affecting Fruit Quality, but with an Increased Yield for Diseased Trees

Insect-protective screen houses can effectively prevent the epidemic of HLB. Region A of orchard 4 was divided into A1 and A2, with A1 under semi-natural conditions in a screen house. In the first survey, the incidence of HLB in these two regions was 4.48% (19/424) and 5.79% (20/345), respectively. Although there were still 424 citrus plants in A1, 22 (5.18%) were affected, with a disease index of 4.36 the next year. By contrast, 38 (11.01%) of the trees were affected in the A2 region, and the disease index was as high as 8.70. Specifically, the severity of HLB and the increased rate of the diseased trees in the insect-proof net area were generally lower than those found in natural conditions. In the two regions from 2019 to 2021, the three-year average HLB incidence was 1.05 ± 0.48% and 6.58 ± 2.85%, respectively. In addition, the tree number reduction in this net-covered region was also lower (0.05% and 0.14% from 2019 to 2020, and from 2020 to 2021, respectively).

Additionally, the fruit quality indexes of both healthy and HLB-affected trees inside (Region A1) and outside (Region A2) the screen house were compared. There was no significant difference in the yield of the healthy trees in these two areas (39.27 ± 7.27 kg for A1 and 33.63 ± 4.89 kg for A2, *p* = 0.149). In contrast, the trees in A1 had a significantly higher crown surface area (103.08 ± 3.84 m^2^ for trees in A1 and 91.09 ± 3.35 m^2^ for trees in A2, *p* = 0.016). By contrast, the yield of the HLB-affected trees in the A1 region was significantly (*p* < 0.001) higher than those found in the A2 region (15.37 ± 0.86 kg/tree and 4.56 ± 0.21 kg/tree, respectively). For the fruits with economic value picked from the two regions, there were no significant differences in fruit quality parameters (i.e., the contents of vitamin C, TSS, TA, TSS/TA ration, and fruit juice rate) and the morpho-physiological parameters (i.e., the FW, FSI, OPT, FF, pericarp rate, and residue rate). However, the red value of the fruit peel outside the screen house (36.96 ± 0.57) was significantly higher (*p* = 0.003) than that of the fruits inside the screen house (32.51 ± 1.02), whereas there was no significant difference noted in the yellow value, color saturation, and brightness of the fruit peel between fruits inside and outside. No *D. citri* was found via the three-year monitoring in the screen house region with the yellow plate method and the knocking method. Collectively, the abundance and types of other citrus pests, as well as their natural enemies, in A1 were less than those found in A2.

### 3.3. Field HLB Incidence Is Influenced by the Direction and Latitude of the Field

#### 3.3.1. Regional Distribution Characteristics of HLB Trees in the Flat Groves

According to the terrain, path, water channels, etc., the orchards were subdivided into multiple regions, as shown in Figure 3. Orchard 1 and orchard 2 are two conventional managed orchards in the Deqing County (Figure 3—1,2), where the citrus is densely planted on flat terrain. Regarding orchard 1, the HLB incidences of the sub-regions in 2019–2020 are shown in Table 2. The geographical position from region A to region C was from north to south, with incidences of HLB decreasing from 14.01% to 3.38% in 2019. Similarly, the incidences in region D to region E also decreased from 4.93% to 2.18%. In 2020, the HLB incidences in the northern segment were still higher than those in the southern segment. Surprisingly, in region B, which was the entrance of the orchard, the incidence increased to the highest recorded of 19.80%. In addition, the incidence also rapidly increased to 17.02% in the center (region D). However, whereas the growth rates of all other regions were higher than 10%, the incidence in region A only increased by 0.65%. Orchard 2 had a similar disease incidence pattern to orchard 1. Region A, located in the southwest, had a three-year average of 16.2% in HLB incidence, lower than those in the other regions. The annual disease growth rate of region D in the northeast was also significantly lower than those found in other regions. The disease incidence growth rates of regions B and C in the center of the orchard were higher than those in the other regions. As the entrance of orchard 2, region B still had the highest annual incidence in the whole orchard.

Although orchard 4 was also built on flat terrain, the practice of comprehensive management was still enacted. ‘Shatangju’, the main cultivar in orchard 4, was planted from region A to region D. However, the pomelo planted in region E was removed in the winter of 2020, due to the serious effects of HLB. There was no doubt that region D, adjacent to region E, had the highest three-year average disease incidence rate (14.8 ± 12.2%) among the whole orchard. Even when the pomelo trees had been completely removed, the HLB incidence of region D increased by 15.89%, whereas other regions did not show a significant increase.

#### 3.3.2. Regional Distribution Characteristics of HLB Trees in the Groves Established along Mountains

Field HLB incidence is influenced by altitude. For orchard 5 and 6, the citrus was planted in both the mountainous and flat areas. Orchard 5 was divided into 12 regions (Figure 3), among which the plot consists of four regions (A, B, C, and D) that were in the southwest direction and had the lowest altitude. Accordingly, the three-year average HLB incidence of these four blocks was relatively higher (2.81 ± 2.67%), and the yearly growth rate was also noted as the highest. Regions G, H, I, and J are the steep slope at the north-east of the orchard, with the most elevated altitude. The highest HLB incidence was found in region J (7.51 ± 6.21%). The terrain of orchard 6 was complex, but generally showed a higher disease incidence in the west than in the east. This orchard was divided into highlands (I, J, and K), slopes (D, E, and F), and lowlands (A, B, and C). The HLB incidences of these three parts had a rising trend from north to south: specifically, region C at 0.63% to region A at 2.45%, region D at 0.38% to region F at 3.41%, and region G at 0.81% to region K at 5.60%. Regarding the survey in 2021, the incidences did not increase significantly when compared to those found in 2020.

### 3.4. Disease Scales Are Related to the Management Measures Intensity

The severity of the diseased trees was found to be related to the management levels. Diseased plants in each comprehensive control orchard were mainly on scale 1, followed by scale 2, and then scale 3. However, the numbers of diseased trees at all three levels in the general management orchard were similar (Table 3) (*p* > 0.05). On the map generated by LocaSpace Viewer 4 (pc) (Figure 3), plants of scale 1 and scale 2 were found to be concentrated, whereas the scale 3 trees were mainly distributed at the edge of the region. Interestingly, there were only a few diseased plants of other grades around these plants of scale 3.

HLB spreading characteristics were generated by three years of data acquired from the orchards. Diseased plants were mainly densely distributed at the edge of the field (e.g., beside the canals and along the roads), but were also sporadically distributed in the center of the block (Figure 3). We speculate that HLB spreads from the edge of the field to the center of different blocks. For example, in orchard 4, regions D and E are at the edge of the field. Diseased trees were clustered in both blocks and were considered as two edge disease centers from where HLB spreads outwards. Likewise, HLB has also spread from the end of the block. The disease trees gathered to the end where most trees were disease-free. For example, in area B, HLB trees were gathered in the north end in 2019. A spread of diseased trees from north to south was found there the following year. Similarly, this also happened in the east of block C.

## 4. Discussion

Yellowing and mottling are two characteristic leaf symptoms that occur after being affected by HLB. In China, the yellowing symptom was more prevalent before the 1960s, whereas mottling became more and more common since the 1970s. This may be explained by cultivar conversion, cultural practice changes, and varied environmental factors [17]. In addition, a shift in the prevalence of CLam to CLas has been observed in citrus orchards in Brazil since 2010 [7,31,32]. These HLB-related shifts (symptoms and pathogen species) could be traced and explained by data generated by yearly surveys, such as in this study of a sufficiently long period, which is also a common means of plant disease epidemiology.

Analyzing the genetic diversity of CLas populations based on polymorphic gene loci will provide important information to guide HLB control, as some CLas strains were found newly imported with the seedlings [33]. In this study, only one CLas was detected in the six orchards. In addition, we previously indicated that the pathogens from these cities were genetically similar based on six gene loci [34]. Here, we specifically analyzed the six CLas populations from the six orchards. CLas strains from the same city were clustered in a bunch. However, although Guangzhou (orchard 5) is next to Huizhou (orchard 4), there is a certain genetic distance between the two CLas populations. Moreover, the CLas populations from Huizhou (orchard 4) and Meizhou (orchard 6) were genetically similar, whereas these two cities were geographically far apart. We suggest that the epidemic of HLB needs to consider more factors, such as the source of the seedlings.

Regardless of the incidence or severity of HLB, the orchards that were implemented with comprehensive methods were in a much better situation than those which were implemented with conventional methods on the premise that the planting scale did not change. This proves the importance of scientific and effective prevention and control methods. Comprehensive management could effectively decrease the rate of novel infection as the average relative control efficacy reached 95.53–99.34% in an assay carried out by Wang et al. [35] for 3 years in Shunchang County, Fujian. Similarly, Yu et al. [36] were involved in an investigation regarding the incidence of citrus HLB in groves under integrated management measures, and they conducted this without any technical measures from 2002 to 2019 in the Zhejiang province. The results show that HLB could be effectively controlled with 6 years of comprehensive management. The continuous removal of HLB trees and the replacement of new trees were noted as the most economical and effective way to control this disease for almost ten years [37]. Nonetheless, even at a time when no affected trees were presented in the orchards, this does not summarily mean that the disease is under control. Instead, it may also be the case that there will be a breakout soon [38]. However, most growers still have no awareness of the scientific prevention and control of this disease. In fact, they still insist on partially or temporarily keeping the infected trees, even if for its only, in actuality, quite limited economic value. There is no doubt that these orchards were destroyed, which resulted in confusion for some other managers and growers with respect to doubting the continuous effectiveness of the ‘three-pronged’ measures (i.e., planting disease-free seedling, timely removing the diseased trees, and killing *D. citri* in the large region) [39]. Consequently, the HLB-affected citruses were not timely removed in orchard 1 and orchard 4 (region E) in this study, thus resulting in the destruction of the orchards. The above facts prove that the ‘three-pronged’ measures are still the most effective prevention and control measures for HLB in recent years. Moreover, a comprehensive quarantine can more effectively control the outbreak of disease [38]. However, as insecticide resistance has a vital negative impact on psyllid control, frequent insecticide application is not recommended. Collectively, constant reproduction and saturated reproduction are of pivotal importance [24].

As one of the new citrus production measures accepted by some growers, citrus under protective screen (CUPS) can efficiently exclude the *D. citri* vector of HLB, thereby producing HLB-free healthy fruits [40]. In actuality, this measure is efficient in insect prevention, as it consequently regulates the epidemics of vector-borne diseases. Ferrarezi et al. [41] also found that screen houses, rather than open-air planting, could also provide a better growing environment for young citruses to accelerate their growth. Moreover, this study also found that the yield of the tree and the economic value of the fruits in the screen house were significantly higher than those found in open-air planting areas. Although the red value of the fruit peel in the screen house was significantly lower than those found in open-air planting areas, which may be due to reduced solar radiation accumulation and greater air temperature [41], there was no obvious difference in the fruit quality and morpho–physiological characteristics. However, CPUS can also alter the microclimate inside the screen house, hence increasing the mite population, and also affecting plant growth to a certain extent [42]. On the ground that the mites were well controlled, the anti-psyllid screen house coverage is suggested to be an acceptable new environmental platform by which to cultivate high-value fresh citrus.

HLB trees usually occur in aggregates or clumps in the field. Furthermore, the direction or within-and-across row effects of HLB appearance have also been noted in China, the Philippines, Reunion Island, and São Paulo [43,44,45,46,47]. Our investigation showed that the occurrence of HLB had a certain regularity in the direction of north to south in the flat land orchard, and the incidence was highest in the north end. Before this study, Gottwald et al. [46] found that the trees infected by CLas tended to gather in the same direction in the field of Shantou, Guangdong. In addition, the occurrence of HLB was influenced by multiple other factors. The natural ACP population was found to be the highest from May to September. During this period, the wind blows from the southeast and southwest. This is relevant due to the fact that *D. citri* passively migrates with the wind. Similarly, the distribution characteristics of HLB-affected trees can be explained by the wind direction. Bassanezi et al. [47] found that the degree of diseased tree aggregation was also positively related to disease incidence in 36 plots from 8 farms in the central region of the São Paulo state. Meng et al. [48] used the aggregation index method to analyze 2900 citrus trees sampled from different locations. They found that the aggregation intensity increased with the rise of the positive rate. In our study, the diseased trees were also clustered at certain points in the different orchards, and most of the clustered diseased trees had the same severity. Once citrus trees were infected with CLas in the orchards, the adult *D. citri* also tended to be clustered there first [49]. This behavior of *D. citri* explains the accumulation of citrus plants that were infected with HLB in the orchards. Most infected trees appeared at the edge of the orchard. Moreover, this phenomenon has been mentioned in some survey articles. Data from China, Reunion Island, Brazil, and Florida all indicated occasional higher-than-expected incidence rates of HLB-positive trees at the periphery of the plantings [43,44,45,46,47,50]. A closer scrutiny of the distribution patterns revealed that the interface of zones with non-citrus crops at its perimeter should be avoided. In addition, planting that was created by roads, canals, ponds, and other features all contribute to HLB epidemics as potential linear and/or curvilinear foci of disease. This is because HLB infections tend to accumulate in proportionally higher incidences at these respective interfaces [46,50,51,52,53,54].

## 5. Conclusions

The distribution and epidemic of HLB in orchards have a certain regularity, which is influenced by the planting environment and conditions, the altitude, orientation, wind direction, varieties of citrus, etc. The question regarding how to fully use comprehensive management in order to curb the spread of the disease will be the key problem in the future. As such, this study provided a reference and basis for the formulation of orchard management strategies for tackling the impact of HLB.

## Figures and Tables

**Figure 1 life-13-00749-f001:**
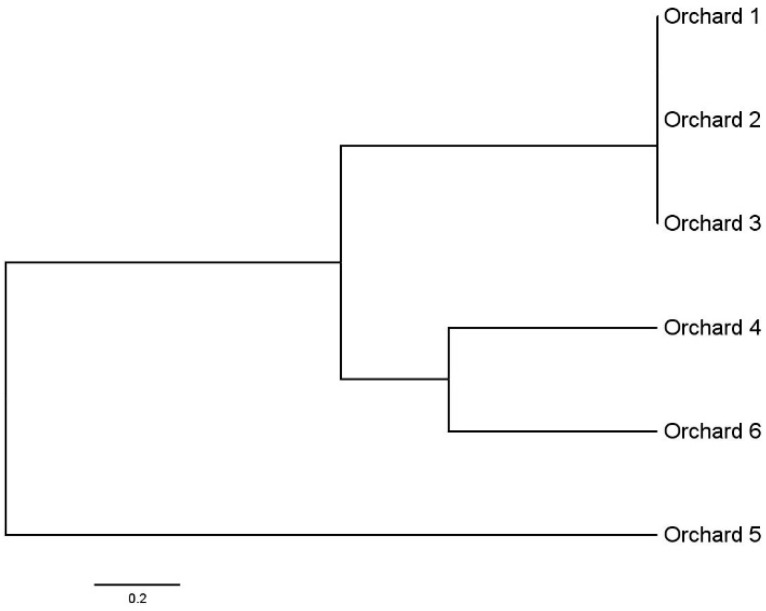
Clustering of *Candidatus* Liberibacter asiaticus population in six orchards of Guangdong Province based on six gene loci. The clustering relationship is based on Nei’s (1972) genetic distance.

**Figure 2 life-13-00749-f002:**
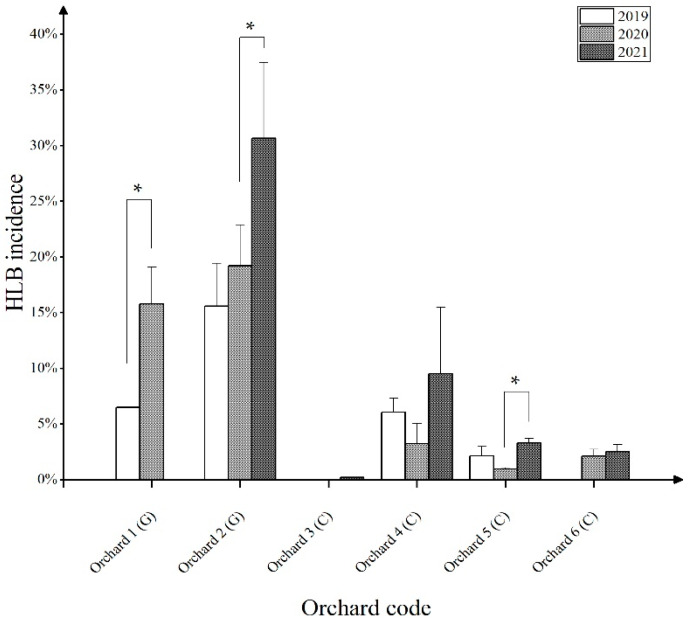
Statistical chart of Huanglongbing (HLB) incidence in the six orchards of the Guangdong Province from 2019 to 2021. G: general management; C: comprehensive control management. The data represent the means ± SD. * *p* < 0.05; (ANOVA with Tukey’s post hoc test).

**Figure 3 life-13-00749-f003:**
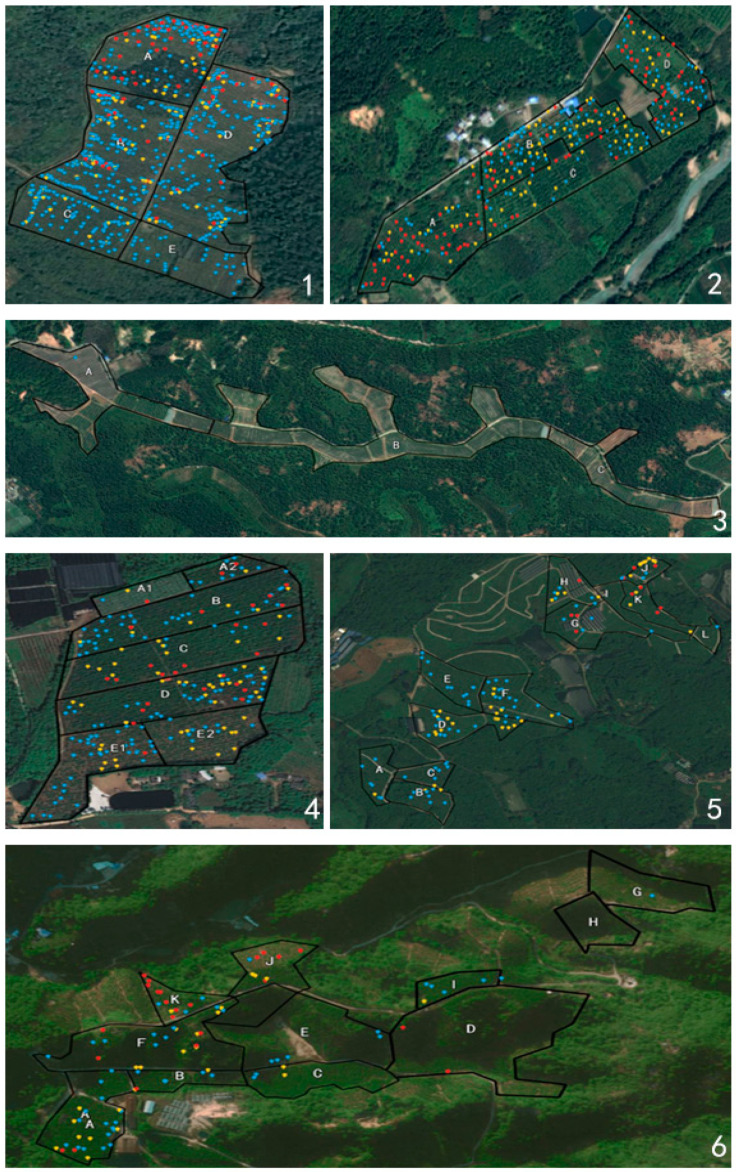
Distribution of the diseased trees in the orchards. The subfigure **1**–**6** correspond to the topographic of orchard 1–6 respectively. Letters A to L were different regions of the groves. The blue dots indicate scale 1 trees, that with 1/3 canopies of each tree affected by HLB. The yellow dots mean scale 2 trees, where more than 1/3 and less than 2/3 canopies of each tree were symptomatic. The red dots represent scale 3 trees, with more than 2/3 canopies affected by HLB. The codes of the groves are in accordance with those in Table 1.

**Table 1 life-13-00749-t001:** Information of the surveyed citrus groves.

Location	Orchard No.	Management Methods	Cultivar	Survey Area (ha)	Geographical Coordinates
Deqing County, Zhaoqing City	1	Conventional management	*C. medica* L. var. *sarcodactylis* Swingle ‘fingered citron’	5	112°12′ E, 23°19′ N
	2	Conventional management	*C. reticulata blanco* var. ‘Gongkan’	10	113°5′ E, 23°13′ N
	3	Comprehensive control	*C. reticulata blanco* var. ‘Gongkan’	20	111°48′ E, 23°15′ N
Boluo County, Huizhou City	4	Conventional management or Comprehensive control	*C. reticulata* Blanco ‘Shatangju’ *C. maxima* ‘Mi Yu’	8	114°28′ E, 23°29′ N
Conghua District, Guangzhou City	5	Comprehensive control	*C. reticulata* Blanco ‘Wokan’	16	113°29′ E, 23°38′ N
Dapu County, Meizhou City	6	Comprehensive control	*C. maxima* ‘Shatian Yu’ and ‘Mi Yu’	10	116°41′ E, 24°22′ N

**Table 2 life-13-00749-t002:** Cumulative annual change in the tree number and regional incidence rate of each orchard.

Orchard and Block	2019		2020			2021			
	Affected Plants	Affection Rate (%)	Affected Plants	Affection Rate (%)	Compare 1	Affected Plants	Affection Rate (%)	Compare 2	Compare 3
1-A	400	14.01	413	14.66	0.65	-	-	-	-
1-B	344	8.04	665	19.8	11.76	-	-	-	-
1-C	142	3.38	524	13.64	10.26	-	-	-	-
1-D	38	2.18	217	11.19	9.01	-	-	-	-
1-E	241	4.93	934	17.02	12.09	-	-	-	-
2-A	105	7.03	291	21.19	14.16	255	20.38	−0.81	13.35
2-B	265	22.23	268	29.32	7.09	409	49.7	20.38	27.47
2-C	227	15.05	151	10.55	−4.5	367	32.97	22.42	17.92
2-D	233	20.58	269	21.3	0.72	258	25.34	4.04	4.76
3-A	-	-	-	-	-	7	0.22	-	-
3-B	-	-	-	-	-	14	0.22	-	-
3-C	-	-	-	-	-	0	0	-	-
4-A1	3	0.76	3	0.79	0.03	6	1.6	0.81	0.84
4-A2	30	8.82	10	3.38	−5.44	20	7.55	4.17	−1.27
4-B	43	2.64	55	3.44	0.8	46	2.94	−0.5	0.3
4-C	89	4.4	29	1.44	−2.96	80	4.38	2.94	−0.02
4-D	180	12.23	59	4.08	−8.15	350	28.09	24.01	15.86
4-E	81	6.88	64	5.82	−1.06	-	-	-	-
5-A	25	2.21	9	1.17	−1.04	16	1.7	0.53	−0.51
5-B	14	1.31	14	1.16	−0.15	77	7.03	5.87	5.72
5-C	7	0.66	7	0.88	0.22	46	4.44	3.56	3.78
5-D	28	1.22	30	1.37	0.15	163	7.93	6.56	6.71
5-E	19	1.15	14	0.84	−0.31	42	2.6	1.76	1.45
5-F	4	0.26	20	1.12	0.86	26	1.67	0.55	1.41
5-G	16	1.41	10	0.84	−0.57	12	1.04	0.2	−0.37
5-H	45	2.57	8	0.57	−2	21	1.43	0.86	−1.14
5-I	21	1.81	1	0.05	−1.76	28	2.62	2.57	0.81
5-J	89	14.59	19	2.98	−11.61	31	4.96	1.98	−9.63
5-K	30	2.56	9	0.77	−1.79	11	1.07	0.3	−1.49
5-L	25	4.17	1	0.26	−3.91	5	1.21	0.95	−2.96
6-A	-	-	27	2.45	-	29	2.57	0.12	-
6-B	-	-	7	1.89	-	11	2.84	0.95	-
6-C	-	-	4	0.63	-	7	1.05	0.42	-
6-D	-	-	3	0.38	-	7	0.86	0.48	-
6-E	-	-	24	2.49	-	25	2.65	0.16	-
6-F	-	-	25	3.41	-	29	3.84	0.43	-
6-G	-	-	1	0.81	-	2	1.57	0.76	-
6-H	-	-	0	0	-	2	0.89	0.89	-
6-I	-	-	6	2.47	-	8	3.42	0.95	-
6-J	-	-	10	2.93	-	14	3.87	0.94	-
6-K	-	-	13	5.6	-	12	4.98	−0.62	-

The orchard number is consistent with Table 1, and the partition number is shown in Figure 3; Compare 1 refers to the difference between the incidence rate in 2019 and 2020; Compare 2 refers to the difference between the incidence rate in 2020 and 2021; and Compare 3 refers to the difference in incidence rate between 2019 and 2021. The dashes mean no deteced diseased trees.

**Table 3 life-13-00749-t003:** Number of HLB-affected plants with different severities in the different orchards.

Orchard Code	Disease Severity Classification			Mean ± SD
	L1	L2	L3	
2	310	276	305	297 ± 18.4 a
4	136	56	28	73.3 ± 56.0 b
5	67	62	13	47.3 ± 30.0 b
6	59	33	28	40 ± 16.6 b
Mean ± SD	143 ± 116.6 a	106.8 ± 113.5 a	93.5 ± 141.2 a	

L1 represents trees with one third of the branches being affected by Huanglongbing; L2 represents trees with more than one-third, but less than two-thirds, of the branches being diseased; L3 means more than two-thirds of the branches of the diseased tree show Huanglongbing symptoms. Values with the small letters, a and b, are significantly different across the line columns. The data represent the means ± SD. ANOVA with Tukey’s post hoc test, at a significance level of 0.05.

## Data Availability

All data are available in the manuscript.

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
