# Peer review of "Current Epidemic Situation and Control Status of Citrus Huanglongbing in Guangdong China: The Space–Time Pattern Analysis of Specific Orchards"

_life, 2023, doi:10.3390/life13030749_

Round 1
Reviewer 1 Report
this manuscript entitled "Current epidemic situation and control status of citrus Huanglongbing in Guangdong China: the space-time pattern analysis of specific orchards" focus on epidemic of CLas in guangdong province in China, and also compare diferent managment in control of citrus HLB. However, this manuscript still have many problems.
introduction is not clear, it is difficult to understand background of this research. In result section, the presenting results can not support to conclusion.
this manuscript is not wroten well, and english need to be improved. So many sentences are viery difficult to understand.
the other problems are in attachment, please check again.
thank you and have a good day.

Author Response
Reviewer 1
this manuscript entitled "Current epidemic situation and control status of citrus Huanglongbing in Guangdong China: the space-time pattern analysis of specific orchards" focus on epidemic of CLas in guangdong province in China, and also compare diferent managment in control of citrus HLB. However, this manuscript still have many problems.
introduction is not clear, it is difficult to understand background of this research. In result section, the presenting results can not support to conclusion.
Response: On behalf of my co-authors, we are very grateful to you for giving us these constructive comments and suggestions on our manuscript. We have rewrote some parts of the introduction, especially the 2nd , the 3rd, and the 5th paragraphs. In the “Results” part, most subtitles were changed and one subtitle was added. The contents of some paragraphs were reformed. For the “Discussion” part, most sentences in the second and third paragraphs were rewrote.
this manuscript is not wroten well, and english need to be improved. So many sentences are viery difficult to understand.
Response: Sorry for the poor language of last edition. After modification according to the comments of all reviewers, we have our manuscript sent to the MDPI for English editing. Hopefully the language of this revised edition is qualified.
the other problems are in attachment, please check again.
Response: We have carefully checked the comments and suggested modifications in the manuscript. Thanks very much for all your valuable comments. We modified or deleted some confusing or unnessary sentences. The edition with modifications was also attached. Please check accordingly.
Reviewer 2 Report
Dear Dr. Deng and Dr. Xu
I reviewed your paper and I think you have to do some more analysis to improve your paper.
1) You need to add a phylogenetic tree to identification of the disease causal agent based on at least 3 genes/gene regions.
2) A photo-plate of infected orchards and fruits is required.
3) In Figure 1 and Table 3 you need to add statistical Tukey Post-hoc comparison.
Regards
Author Response
Reviewer 2
I reviewed your paper and I think you have to do some more analysis to improve your paper.
Response: Thank you very much for taking your time to review this manuscript and for your valuable comments. We have tried our best to improve the manuscript.
1) You need to add a phylogenetic tree to identification of the disease causal agent based on at least 3 genes/gene regions.
Response: Special thanks to you for your good comments. Before carrying out the investigations mentioned in this study, we have performed a province-wide study to explore the intraspecific genetic structure and genetic diversity of 176 ‘Candidatus Liberibacter asiaticus’ isolatess based on six polymorphic gene loci. Results showed that the CLas population in the cities mentioned in this study were similar (For details, please refer to the Figure 3 and Figure 4 in “Huang et al., Population diversity of "Candidatus Liberibacter asiaticus" in Guangdong Province based on different gene loci. J South China Agri Uni 2020, 41(2), 66-75. doi: 10.7671/j.issn.1001-411X.201907007”). Based on this, we started this study. We have added this background in the part of “2.1. Orchard information and survey methods”.
2) A photo-plate of infected orchards and fruits is required.
Response: Accepted. Thanks. Please refer to Figure A1.
3) In Figure 1 and Table 3 you need to add statistical Tukey Post-hoc comparison.
Response: Accepted and added. Thanks.
Reviewer 3 Report
The manuscript entitled "Current epidemic situation and control status of citrus Huanglongbing in Guangdong China: the space-time pattern analysis of specific orchards" sets out to provide guidance for the early warning of citrus greening or Huanglongbing (HLB), enhance the prevention awareness of growers, and give suggestions on the specific effective measures for the prevention and control of HLB in areas where HLB is severely endemic. The study is on a topic of relevance and general interest to the journal's readers. However, I have several concerns about presenting the data that should be addressed before publication.
· The authors need to carefully read the manuscript to correct typos and grammar to improve the manuscript. Also, use one font type throughout the manuscript.
· Any abbreviation must be associated with the full name at the first mention in the manuscript
· In the material and methods section, any chemical, equipment, or software must have the complete source, including company, city, state, and country
· I Table 2 indicates what do you mean by the dashes, is it means not measured, not detected?
Author Response
Reviewer 3
The manuscript entitled "Current epidemic situation and control status of citrus Huanglongbing in Guangdong China: the space-time pattern analysis of specific orchards" sets out to provide guidance for the early warning of citrus greening or Huanglongbing (HLB), enhance the prevention awareness of growers, and give suggestions on the specific effective measures for the prevention and control of HLB in areas where HLB is severely endemic. The study is on a topic of relevance and general interest to the journal's readers. However, I have several concerns about presenting the data that should be addressed before publication.
Response: Thank you for handling our manuscript and for your positive evaluation on it. All your constructive comments on the manuscript have been carefully considered. We tried our best to revise our manuscript according to the comments. The version with changes tracked is also uploaded. Some revision notes, point-to-point, are given as follows:
- The authors need to carefully read the manuscript to correct typos and grammar to improve the manuscript. Also, use one font type throughout the manuscript.
Response: Accepted and revised. We have worked on both language and readability. The language of the manuscript has also been edited by the MDPI.
- Any abbreviation must be associated with the full name at the first mention in the manuscript
Response: Thanks. We have checked all the abbreviations and revised when necessary.
- In the material and methods section, any chemical, equipment, or software must have the complete source, including company, city, state, and country
Response: Accepted. We thank you for pointing out this issue. We have double-checked all the related sentences and revised accordingly.
- I Table 2 indicates what do you mean by the dashes, is it means not measured, not detected?
Response: Accepted. We have added “The dashes mean no deteced diseased trees”. Thanks.
Round 2
Reviewer 1 Report
This manuscript is improved a lot. Now it is better than previous one.
Author Response
Dear reviewer,
Thank you for handling our manuscript and for all your constructive comments on it. We have tried our best to revise our manuscript according to all the reviewers comments. The version with changes tracked was also uploaded.
Hopefully this edition will meet the requirement of the journal.
Best regards,
Meirong Xu
Reviewer 2 Report
Dear Corresponding Author
As I suggest you, it is needed to add a phylogenetic tree. It is very clear.
Regards
Author Response
Dear reviewer,
We are very grateful to you for all your constructive comments and suggestions on our manuscript.
We have add the cluster analysis of “Candidatus Liberibacter asiaticus” populations in the orchards based on six polymorphic gene loci. The phylogenetic tree was shown in Figure 1. Related content has also been added in the “Materials and methods”, “Results”, and the “Discussion”. Please find the resubmitted files.
Thanks a lot!
Best regards,
Meirong Xu